# Review of the Literature on Partial Resections of the Gallbladder, 1898–2022: The Outline of the Conception of Subtotal Cholecystectomy and a Suggestion to Use the Terms ‘Subtotal Open-Tract Cholecystectomy’ and ‘Subtotal Closed-Tract Cholecystectomy’

**DOI:** 10.3390/jcm12031230

**Published:** 2023-02-03

**Authors:** Raimundas Lunevicius

**Affiliations:** Department of General Surgery, Liverpool University Hospitals NHS Foundation Trust, Lower Lane, Liverpool L9 7AL, UK; raimundas.lunevicius@liverpoolft.nhs.uk; Tel.: +44-7528044888; Fax: +44-1515298547

**Keywords:** gallbladder surgery, history, cholecystectomy, partial cholecystectomy, subtotal cholecystectomy, trends, classification, conception

## Abstract

Current descriptions of the history of subtotal cholecystectomy require more details and accuracy. This study presented a narrative review of the articles on partial resections of the gallbladder published between 1898 and 2022. The Scale for the Assessment of Narrative Review Articles items guided the style and content of this paper. The systematic literature search yielded 165 publications. Of them, 27 were published between 1898 and 1984. The evolution of the partial resections of the gallbladder began in the last decade of the 19th century when Kehr and Mayo performed them. The technique of partial resection of the gallbladder leaving the hepatic wall in situ was well known in the 3rd and 4th decades of the 20th century. In 1931, Estes emphasised the term ‘partial cholecystectomy’. In 1947, Morse and Barb introduced the term ‘subtotal cholecystectomy’. Madding and Farrow popularised it in 1955–1959. Bornman and Terblanche revitalised it in 1985. This term became dominant in 2014. From a subtotal cholecystectomy technical execution perspective, it is either a single-stage (when it includes only the resectional component) or two-stage (when it also entails closure of the remnant of the gallbladder or cystic duct) operation. Recent papers on classifications of partial resections of the gallbladder indicate the extent of gallbladder resection. Subtotal cholecystectomy is an umbrella term for incomplete cholecystectomies. ‘Subtotal open-tract cholecystectomy’ and ‘subtotal closed-tract cholecystectomy’ are terms that characterise the type of completion of subtotal cholecystectomy.

## 1. Introduction

The first cholecystectomy was performed by Carl Langenbuch in the Lazarus hospital of Berlin on 15 July 1882 [1]. It was documented as a case of extirpation of the gallbladder for chronic cholecystitis in a 43-year-old male patient. Since then, cholecystectomy has gradually become a standard treatment paradigm for symptomatic or complicated gallbladder stone disease. However, the unpredictable course of the inflammatory process and pericholecystic fibrosis, frequently hampered by biliary infection, variations of ductal and vascular anatomy, liver cirrhosis, adhesions related to previous surgeries, the poor physiological condition of the patient, and even limited experience in gallbladder surgery predispose the specific situations during gallbladder operation, which are referred to as extraordinary, high-risk, dangerous or, simply difficult. In such circumstances, which are common [2,3,4], the rescuable (viz., damage control) surgical procedures should immediately be considered because they are the only reliable method to complete the operation safely. Subtotal cholecystectomy is one of them.

It is noteworthy to mention the remarkable history of another damage control surgical procedure—cholecystostomy. In 1859, a London internist, John Thudichum, encouraged surgeons to perform a gallstone operation that involved fixing the gallbladder to the abdominal wall through a small incision and then, having allowed adhesions to form, incise it to extract the gallstones, leaving the resultant fistula to heal spontaneously [5]. However, the first successful gallbladder drainage procedure, named lithotomy of the gallbladder, was performed on a 30-year-old female patient by John Stough Bobbs in Indiana in 1867. In 1878, a decade later, James Marion Sims (South Carolina) introduced the term ‘cholecystostomy’ [6].

With this in mind, one can assume that the history of the subtotal cholecystectomy operation is even more complicated. Therefore, it is disputable regarding the nomenclature and classifications of the variants of resections of the portions of the gallbladder and types of completion of this procedure. Indeed, after analysing materials published previously [7,8,9], it turned out that the current descriptions of the history of subtotal cholecystectomy required more details, and the information about the chronological evolution of subtotal cholecystectomy often warranted to be more accurate.

This study presented a narrative review of the papers on limited resections of the gallbladder published between 1898 and 2022, focusing on its technical aspects, nomenclature, and classifications. With this, the study also purposed to acknowledge the contributions of those general surgeons who developed the subtotal cholecystectomy theory and practise. Furthermore, this article aimed to assess the recent trends in subtotal cholecystectomy and revise the terms currently used to characterise the type of completion of this surgical procedure.

## 2. Materials and Methods

The material for the paper was prepared, and the manuscript was drafted according to the six items of the Scale for the Assessment of Narrative Review Articles [10]. The items are as follows: explanation of the review’s importance, statement of the concrete aims, description of the literature search, referencing, scientific reasoning, and presentation of relevant and appropriate presentation of data.

The primary literature sources were identified using a cascade mechanism. First, MEDLINE/PubMed, Embase, and Cochrane bibliographic databases and Google Scholar were used to identify and obtain papers on limited cholecystectomy published between 1985 and 2021. The search technique, including a specific search string, was described previously [7,9]. Second, the generic search technique using the keywords ‘partial cholecystectomy’ and ‘subtotal cholecystectomy’ was employed to identify papers published before 1985. Third, the lists of references of all obtained articles were scrutinised to determine other historical literature sources. When reports were unavailable via bibliographic databases, the Royal College of Surgeons of England (and, subsequently, the British Library) were further contact points. Furthermore, the author’s private networks in Argentina and Spain were communicated. 

In this paper, the terms ‘limited resection of the gallbladder’, ‘limited cholecystectomy’, ‘partial resection of the gallbladder’, ‘partial excision of the gallbladder’, ‘less-than-total cholecystectomy’, ‘incomplete cholecystectomy’, ‘compromise cholecystectomy’, and ‘partial cholecystectomy’ are synonymous. The remaining part of the resected gallbladder was defined as a gallbladder remnant.

Notably, aspiration of the liquid content of the gallbladder, incision and evacuation of calculi and debris from the cavity of the gallbladder, surgical instruments used, chemical or mechanical treatment of the mucosa of the remnant of the gallbladder, the number of drains utilised, and the features of the general anaesthetic were not the objects of this review. Moreover, the use of jargonic terms was minimised in this paper. For example, the anatomic terms ‘hepatic and peritoneal walls of the gallbladder’ were used instead of ‘posterior and anterior walls of the gallbladder’, with one exception—the quotations of the historical papers were not edited.

The data-based figures were generated using GraphPad Prism version 9.5.0 (525) for macOS (GraphPad Software, Inc., San Diego, CA, USA).

## 3. Results

### 3.1. General Overview

The literature search yielded 165 publications. Of these, 27 items were published between 1898 and 1984 [11,12,13,14,15,16,17,18,19,20,21,22,23,24,25,26,27,28,29,30,31,32,33,34,35,36,37]. The other 138 full-text papers on at least five operations were published between 1985 and 2022, as shown in Figure 1 [38,39,40,41,42,43,44,45,46,47,48,49,50,51,52,53,54,55,56,57,58,59,60,61,62,63,64,65,66,67,68,69,70,71,72,73,74,75,76,77,78,79,80,81,82,83,84,85,86,87,88,89,90,91,92,93,94,95,96,97,98,99,100,101,102,103,104,105,106,107,108,109,110,111,112,113,114,115,116,117,118,119,120,121,122,123,124,125,126,127,128,129,130,131,132,133,134,135,136,137,138,139,140,141,142,143,144,145,146,147,148,149,150,151,152,153,154,155]. 

### 3.2. First Described Operations, 1898–1900: Hans Kehr, William J. Mayo, and Bertram C. Stevens

Accessible primary literature sources indicate that Kehr [11] and Mayo [12], being thousands of miles away from each other in Halberstadt and Rochester, were the first surgeons who performed and described a partial resection of the gallbladder in difficult circumstances. 

In 1901, William W. Seymour, a surgeon and translator, published H. Kehr’s book, which contained two case reports on partial gallbladder excision for gallstones [11,156]. The first patient was a 27-year-old female who underwent an operation named ‘Resection of the gallbladder’ due to recurrent acute cholecystitis with a mass of inflammatory origin and chronic pancreatitis on 17 February 1898. Notably, the term ‘resection’ means excision of the part of the organ (gallbladder, in the context of this paper) by German medical terminology. The quote from the book [11] (p. 194):


*‘The gallbladder is not visible; it is intimately adherent to the inflamed omentum. It is possible only with difficulty to free the gallbladder, which is further adherent to the stomach and the greatest part of the posterior surface of the duodenum. In so doing, its thickened and soft wall tears. There appear in view a number of small to pea-sized roundish yellow stones with thick pus… The stones were removed with forceps. One intends to extirpate the gallbladder but finds the adhesions on the posterior surface separable only with great difficulty; besides, it is also evident that perforations have occurred and stones will lie behind the bladder in the adhesions; the removal of these is very difficult; on this account, one removes so much of the gallbladder wall that only in fact hardened posterior wall and the part of the bladder lying next to the cystic duct remains. With this, severe bleeding occurs from the cystic artery, which is controlled by ligature. Now there yet stick two stones in the cystic duct, which are removed with great difficulty. Then the bladder is sewn upon itself, some omental bands are ligated, a trip of gauze introduced down to the sutures, and the abdominal wall closed…’.*


In current terms, it is a description of subtotal cholecystectomy with the closure of the remnant of the gallbladder in a surgical situation when neither cholecystostomy (a common operation in that historical period) nor total cholecystectomy was possible. The material provided suggests that H. Kehr left the entire hepatic wall in situ; however, the description of the second operation performed on the same patient on 24th May 1898 due to persistent extracorporeal bile leakage leads the reader to believe that H. Kehr detached the distal portion of the hepatic wall of the gallbladder from its anatomical site during the first operation, excised the part of the gallbladder circumferentially and sutured the stump of the remnant gallbladder. H. Kehr stated after the second operation [11] (p. 195):


*‘Separation of the stomach from the peritoneum; in so doing, the stomach tears, partial suture of the opening. The gallbladder stump is put in anastomosis with the stomach at the point of the tear in the stomach. Cystico-gastrostomy’.*


On 5th December 1898, the second resection of the gallbladder for chronic empyema of the perforated gallbladder with adherent omentum around it was performed by Kehr on a 60-year-old female patient. The operation was titled ‘atypical ectomy’ (i.e., atypical cholecystectomy). A quote describing the resection is concise [11] (p. 220): 


*‘Excision of the very fragile gallbladder at the level of the neck. Introduction of the tube into the stump, which is firmly sutured’.*


Since this description of the operation is also imprecise, it is difficult to attribute this operation to one or another technical type or variant of limited cholecystectomy. However, this description shows that a portion of the gallbladder being distally from its neck was removed (the equivalent of his first resection of the gallbladder). Notably, the extracorporeal bile leak developed postoperatively. However, the fistula firmly closed in approximately 5 weeks. The patient was discharged from the hospital on 22nd January 1899.

In 1899, W.J. Mayo reported that 132 operations on the gallbladder and bile ducts were performed during the 9 years at St. Mary’s Hospital, Rochester [12]. Unfortunately, their dates were not provided to the readers. Two techniques of limited resection for non-malignant gallbladder disease were described. The first method was used for three patients. It entailed opening the gallbladder, removing the mucous membrane while leaving the cystic duct open, and suturing the muscular and peritoneal layers of the gallbladder into the incision. Therefore, it is difficult to admit that a total mucosectomy of the gallbladder is a partial cholecystectomy or modification. However, a total mucosectomy of the gallbladder for advanced acute cholecystitis is a resectional surgical procedure. H. P. Ritchie named it subserous cholecystectomy [23]. 

W.J. Mayo also reported that he excised the gallbladder and covered the bleeding surface of the attached liver and the infected stump with gauze held firmly in position by sutures of fine catgut in three other patients with acute phlegmonous and gangrenous cholecystitis [12]. He did not specify whether the infected stump was a proximal portion of the gallbladder; however, the logic of the description dictates that this was the case. Therefore, partial circumferential resection of the gallbladder with the closure of the stump of the gallbladder was a recognised operation in St. Mary’s Hospital, Rochester, within the last decade of the 19th century. 

In 1900, partial circumferential cholecystectomy with surrounding liver parenchyma was also performed due to obvious complicated calculous cholecystitis and suspected malignant gallbladder disease. Bertram C. Stevens, in 1901, described a case of the combined procedure—cholecystectomy, hepatectomy, and pylorectomy—performed in Leeds on 9th August 1900 by Mr. Mayo Robson for gallbladder calculi, empyema and cancer with a fistula between the gallbladder and the pyloric end of the stomach, through which the gallbladder empyema was discharging into the stomach. B. C. Stevens wrote [13] (p. 879): 


*‘A V-shaped portion of the margin of the liver and most of the gall bladder were removed, only a small portion of apparently healthy gall bladder being left… A tube was placed in the cystic duct, and what was left of the gall bladder was sewn firmly around the tube’.*


This case is relevant to this review in four areas. First, this is an example of a challenging surgical situation associated with acute calculous cholecystitis, gallbladder empyema, cancer, and cholecystopyloric fistula. Second, it demonstrates how the surgeon should be prepared for any contingency when operating on complicated gallstone disease. Third, it is a circumferential removal of a significant portion of the gallbladder with the surrounding liver parenchyma. Fourth, the closure of the remnant of the gallbladder with the formation of controlled tube cholecystostomy.

Briefly, the juncture of the 19th and 20th centuries was marked by the advent of a new operation—a partial resection of the gallbladder—to avoid dangerous intraoperative complications in difficult surgical circumstances when traditional cholecystostomy and cholecystectomy were not the options. In the future, this surgical gallbladder procedure will be named partial cholecystectomy.

### 3.3. Limited Resection of the Gallbladder to Reduce Mortality, 1920–1930

It should be acknowledged that, at the beginning of the 20th century and pre-antibiotic era, the general surgeons-innovators aimed to reduce the mortality from sepsis secondary to suppurative or gangrenous cholecystitis performing simple gallbladder surgical drainage via a longitudinal incision through the entire length of the gallbladder peritonealised wall without, in most of the cases, or with, in some cases, resection of this wall. Notably, mortality from biliary sepsis in some hospitals was approximately 50%, according to a brief overview provided by W. L. Estes (V. Pauchet paper, 1924, referenced) [16,17]. Descriptions of partial resection of the gallbladder peritoneal wall without or with ligation of the cystic duct (also, without or with drainage of this duct) are found in other papers published between 1920 and 1930 [14]. The split of the gallbladder from the fundus to the cystic duct, removal of stones, resection of the peritoneal wall, leaving the whole hepatic wall attached to the liver, curettage of the mucosa of the remnant gallbladder, and the application of chemicals on it were the key elements of this operation according to Martin’s remarks [14] and Estes’ overviews [16,17], which referenced Bengolea (1920), De Martel (1923), Zabala (1924), Pauchet (1924), Gatch (1927), Zimmerman (1927), and Haggard (1930) papers. Notably, De Martel’s (Paris, France) paper, published in 1923, described the division of the cystic duct before cutting away the peritoneal wall of the gallbladder. 

In 1926, E. Denegre Martin (New Orleans, Louisiana) described eight life-saving gallbladder operations, of which three—the sixth, the seventh and the eighth—fitted the criteria of limited cholecystectomy [14]. All three descriptions are unique as they highlight the relationship between the excision of the peritoneal wall of the gallbladder, open cystic duct and unpredictable postoperative course due to bile leakage. The following is a quote from a description of the 6th operation performed in July 1924 [14] (p. 199): 


*‘The gallbladder was dark blue and greatly distended but free of adhesions. As we attempted to lift it into view, it ruptured, revealing the fact that the tissues were exceedingly friable. An incision was made through its entire length, the redundant tissues removed, and bleeding points ligated. As no bile was escaping through the cystic duct, a cigar drain with the tube in the centre was sutured against the remaining mucous surface of the gall bladder. Twenty-four hours later, bile began to flow’.*


The 7th description is the most representative as it emphasises the extent of removal of the peritoneal wall of the gallbladder and the importance of double drainage when bile leak, as a postoperative event, is real [14] (p. 200): 


*‘The fundus was incised, a number of small stones were removed, and the mucosa dried and swabbed out with iodine. It was then split from the fundus to the cystic duct, and the redundant wall cut away almost to the liver attachment. A cigar drain was inserted and sutured in position; another was placed in the pouch below. This case has recovered from an operation, and I am very confident it will have no recurrence’.*


E.D. Martin concluded that if the gallbladder is greatly distended, as a significant portion of the gallbladder wall as is considered necessary is excised, and all bleeding points are ligated, the chance to survive should be provided to the patient. However, if the gallbladder is small, no effort should be made to remove any redundant tissue. Therefore, E.D. Martin described the surgical technique of the limited resection of the gallbladder in the form of excision of the peritoneal wall, its indication, and contraindication [14]. 

Eleven years later, in 1937, W. D. Haggard from Nashville (Tennessee), discussing Frank H. Lahey’s paper on strictures of the common hepatic and bile ducts, provided commendation on E.D. Martin’s operation for the most severe cases of gangrene of the gallbladder, which provide the greatest danger of accidental injury to the bile ducts [157]. Haggard emphasised that it is an excellent method to accomplish difficult cholecystectomy. The comment also stressed that D. Gatch independently performed the same operation. 

### 3.4. Introduction of the Term ‘Partial Cholecystectomy’, 1931: The Series of W. L. Estes Papers

In 1931, William L. Estes provided a detailed description of the operation for acute calculous gangrenous cholecystitis with induration of the cystic and common ducts, which was performed in 1929, and named ‘partial cholecystectomy’ [15,16,17]. Seven cases were described regarding the history of the biliary disease, examination results, operation features, and postoperative course. Five summary points by W. L. Estes include the following: (1) although complete cholecystectomy is indicated, it is technically inadvisable and probably dangerous to perform in the event of inflammatory induration of the cystic duct; (2) partial cholecystectomy is an operation of choice in such circumstances; (3) the part of the gallbladder that is attached to the liver is allowed to remain; (4) the cut surface of the gallbladder is sutured by lock stitch; (5) partial cholecystectomy is not a new procedure. W. L. Estes wrote [15] (pp. 119–120): 


*‘A short time later, I was confronted by a case of gangrenous cholecystitis with stones in which complete cholecystectomy was definitely indicated; because of inflammatory induration about the cystic duct; however, this method seemed technically inadvisable and probably dangerous. I therefore resorted to this same splitting of the gallbladder after removal of the stones but supplemented it by trimming off the excess portion of the gallbladder close to the fossa in the liver, attempting to obtain the effect of a complete cholecystectomy to permit drainage of the cystic duct and to avoid a two-stage operation, as is often necessary when only cholecystostomy is done. This partial cholecystectomy I have used in seven carefully selected cases. Convalescence has usually been uneventful; there have been no mortality and no evidence of peritonitis.’*


In 1938, W.L. Estes published two reports on 48 partial cholecystectomy cases, where a split of the peritoneal wall of the gallbladder with scissors down to within 1–2 cm of the cystic duct was clarified [16,17]. The application of a ligature to control the bleeding from the cut edges or continuous lock stitches on each side were considered technical alternatives of the same value. Additionally, Thorek’s method of destroying the mucosa of the remnant of the gallbladder by bipolar diathermy [18,19,20] has been acknowledged as being applicable in partial cholecystectomies. 

Partial cholecystectomy was considered as a factor that decreased postoperative mortality (2.1%, one death reported), with long-term follow-up results comparable with complete cholecystectomy. Furthermore, a follow-up examination of 42 (89.4%) patients conducted within 14 years after partial cholecystectomies using the same template assessed five categories of postoperative sequelae—relief from gallbladder disease symptoms, subsequent involvement of the common duct, the fate of the remnant of gallbladder allowed to remain, postoperative hernia, and miscellaneous—and was a remarkable feature of these studies on the long-term clinical outcomes of partial cholecystectomy. 

Two broad indications—first, acute suppurative or gangrenous cholecystitis or empyema of the gallbladder, mainly when induration exists about the cystic and common ducts. Second, a small, densely adherent, thickened, atrophic gallbladder, which cannot be easily separated from the bed of the liver, was described by the statement that ‘*in no way should partial cholecystectomy be considered to supplant complete cholecystectomy when complete removal can be safely accomplished*’ [17] (p. 854).

Finally, the abstract of the discussion between Dr. William D. Haggard (Nashville, Tennessee), Dr. Donald Guthrie (Sayre, Pennsylvania), Dr. Moses Behrend (Philadelphia), and Dr. William L. Estes is a testimony of the opposite schools of thought within the third and fourth decades of the 20th century [17]. The following seven points were highlighted. First, Dr. Estes’ operation is the correct one to employ in a desperate situation. Second, when the isolation of the cystic duct is difficult, removal of the gallbladder entails a considerable risk of injury to the deeper ducts. Third, poor exposure of the gallbladder, improper mobilisation of the infundibulum, and haemorrhage from the cystic artery are other frequent causes of injury to the common duct, with the development of a postoperative biliary fistula or a stricture of the common duct. Fourth, the removal of mucosa from the remnant of a gallbladder is the element of this operation. Fifth, a closure of the biliary tract via sewing the cut edges of the remnant of the gallbladder over a catheter in the cystic duct is possible. Sixth, perihepatic drainage should be adequate. Seventh, compromise cholecystectomy is another name for partial resection of the gallbladder.

### 3.5. H. P. Ritchie, 1937: A Technique of Cholecystectomy for the Complicated Case of Gallbladder Disease 

Harry P. Ritchie (St. Paul, MN, USA) effectively overviewed the historical tensions on high mortality and the genesis of the conception of partial cholecystectomy [23]. In the first decade of the 20th century, H. P. Ritchie wrote that there was an active discussion on the relative merits of cholecystotomy, cholecystostomy and cholecystectomy reducing postoperative mortality (range 2–25%) associated with severe forms of gallbladder disease and its complications. Arthur William Mayo-Robson (Leeds, London, UK), Berkeley George Andrew Moynihan (Leeds), John Blair Deaver (Philadelphia, PA, USA), Maurice Howe Richardson (Harvard, MA, USA), John Benjamin Murphy (Chicago, IL, USA), and William James Mayo (Rochester, MN, USA) were active discussants. The possible increase in mortality following cholecystectomy was one of the topics of discussion among these distinguished surgeons of that time. H.P. Ritchie stated that Moynihan categorically suggested a careful selection of patients for cholecystectomies and highlighted that post-cholecystectomy mortality should be no larger than that in cholecystostomy. This discussion, which affected the real-life criteria globally at that time, provided valid reasons for clinicians to consider general patient factors and other technical details of and indications for limited operation on the gallbladder. This discussion also helped formulate an emergency plan for difficult and dangerous surgical circumstances related to severe inflammatory changes around and within the hepatoduodenal ligament. The advent of a surgical procedure, which entails resection of a part of the gallbladder, was inevitable, based on the line of thought in H. P. Ritchie’s paper.

Summarising 16 partial cholecystectomies and the available literature, H.P. Ritchie stressed that there are conditions ‘*where the attempt at formal excision carries with it a risk of an operative misstep, a stormy convalescence, or a fatality*’ [23] (p. 582). These conditions went beyond gross gallbladder pathology and were presented in this order: the obese patient, incomplete exposure, inadequate relaxation of the fighting anaesthesia, the aged and debilitated patient, and hypertension. A formulation of the general indication for partial cholecystectomy was precise. The quote [23] (p. 582):


*‘When the landmarks are clouded, when the excision carries a risk of injury to the structures about the gallbladder, when the integrity of the gallbladder wall is uncertain’.*


H.P. Ritchie’s report on a four-step technique of gallbladder resection, leaving a part of the gallbladder in situ, and three objections to such a surgical procedure are rich with specific phrases, terms, and comparisons of the situations to some common resemblances. The examples include: difficult and dangerous circumstances, the most extraordinary and adverse conditions, the definitive margin of safety (performing the longitudinal split of the wall of the gallbladder), the judgment of the operator, the ‘wing’ of the gallbladder (splitting it by two opposed symmetrical incisions), a strip of the gallbladder wall and mucous membrane significantly wide to preserve the normal attachments to the liver, which resembles a ladle, the handle of which is the strip on the liver, and the cup is the mucous membrane lining the base of the gallbladder, the re-formation of the gallbladder [158], a plan of an emergency nature, the risks of injury to the common duct, no damage to structures outside the field.

### 3.6. Partial Cholecystectomy, Middle of the 20th Century: The Trend in Surgery for Difficult Gallbladder

A few studies indicate that a partial cholecystectomy in the form of W. L. Estes’ description was a recognised surgical practise [21,22,27]. E. Starr Judd and J. Roberts Philips (Rochester, MN, USA) emphasised that a partial cholecystectomy is preferable to cholecystostomy while operating on a patient with acute cholecystitis [21]. Reporting results of 508 consecutive cases of acute cholecystitis, they stated that gallbladder removal, except the portion embedded in the liver, was performed on 149 of their patients. Cholecystostomy was performed in 89 patients. Furthermore, they emphasised that if cholecystectomy is performed in a gallbladder that was shrivelled, almost destroyed, due to the inflammatory process, every effort should be made not to traumatise the surface of the liver. They recommended leaving a sufficient amount of the fibrous wall of the gallbladder on the surface of the liver to prevent the extension of the disease into the hepatic parenchyma.

James McKenty (Winnipeg, MB, Canada) reported 30 partial cholecystectomies with only one fatality within the series of 76 cholecystectomies (viz. 46 were complete cholecystectomies) and defined the procedure in this manner [22] (p. 239):

‘*A partial resection of the gallbladder, leaving the portion attached to the liver undisturbed, is, in the presence of sepsis, a safer procedure than complete cholecystectomy, which leaves a denuded, raw gallbladder bed on the liver surface, which is prone to absorb the toxins in the region*’.

Arthur I. Lerner’s (Winnipeg, MB, Canada) remarks on partial cholecystectomy, a conventional excision of the peritoneal wall of the gallbladder, leaving the hepatic wall attached to the liver without a ligature of the cystic duct, are useful in a few instances [30]. First, no exposure to the cystic duct opening is necessary, provided the stones are not present in the common duct. Second, there is no danger of injuring important bile ducts and vessels, for no dissection to expose the cystic duct is needed. Third, place one Penrose drain approximately 1 inch wide instead of multiple drains. Fourth, extracorporeal bile leakage is not undesirable. Fifth, if the main flow of bile through the common duct is unimpeded, the biliary fistula closes. 

Consequently, in 1952, R. J. McNeill Love (London, UK) emphasised that a partial cholecystectomy was one of the modern trends in biliary surgery [24] (p. 214): 


*‘This procedure is occasionally useful if the gallbladder is difficult to access, especially if it tends to be buried in the liver. Additionally, in cases of acute cholecystitis with partial gangrene of the gallbladder, removal of the free portion with coagulation of the part adherent to the liver is the method of choice’.*


### 3.7. Substitutive Methods of Cholecystectomy: Adding the Value to Partial Cholecystectomy Theory

Max Thorek (Chicago, IL, USA) in 1936 and William A. McElmoyle (Victoria, BC, Canada) in 1954 described the methods of modified cholecystectomy [18,31]. The first one was named ‘electrosurgical obliteration of the gallbladder without drainage’ (briefly, cholecystelectrocoagulectomy) and viewed as a modification of complete cholecystectomy. Therefore, it was recommended and used routinely without discrimination between complex and straightforward gallbladders. The second one was shown as a modification or technique of cholecystectomy for the difficult gallbladder [31]. 

#### 3.7.1. Cholecystectomy by M. Thorek

Thorek’s operation [18] should be viewed considering Bruno Oskar Pribram’s (Berlin, Germany) electrosurgical operation for gallbladder diseases, which was introduced into practise in 1922 [32,33]. The Pribram’s method entailed a division of the cystic duct and artery, wide longitudinal cholecystotomy via the peritoneal wall, mucoclasis, and suturing of the folded serosa flaps of the opened gallbladder. Devitalisation of the mucosa of the slitted gallbladder using monopolar diathermy carbonising effect (viz. fulguration) on tissues was the essence of Pribram’s operation. Therefore, the operation of Pribram has a minimal link with partial cholecystectomy, as no full-thickness excision of any part of the gallbladder existed.

M. Thorek adapted Pribram’s method in 1933 and modified it after numerous electrosurgical experiments presented as a thesis in 1936 [18]. The final version of the conception of Thorek’s gallbladder surgery was published in 1954 [20].

Thorek’s cholecystelectrocoagulectomy without drainage entails the emptying of the gallbladder by aspiration and expression of gallstones through an incision in the fundus, exposure of the common ducts via opening cholecystoduodenal ligament so that the cystic duct and artery can be ligated with safety, the longitudinal opening of the gallbladder, removal of the peritoneal wall of it, electrocoagulation of the rest of the mucosa of the gallbladder using an electrode of bipolar current to a tissue surface, approximation of the electrocoagulated edges of the gallbladder by a few interrupted sutures or one continuous suture, and additionally, suturing of the section of omentum over the coagulated hepatic wall of the gallbladder [18]. Notably, Thorek and the advocates of the operation proved that electrosurgical obliteration of the gallbladder obviating the necessity for drainage was carrying a smaller risk of postoperative complications, considerable less postoperative pain, shortened convalescence, and a significant reduction in mortality rate [18,20,26,29]. 

M. Thorek did not use the terms ‘partial cholecystectomy’ and ‘subtotal cholecystectomy’. However, M. Thorek introduced the term ‘a secure sterile tampon’, which emphasises the remaining fibrosed muscular layer of the hepatic wall of the gallbladder. The phrase ‘*a secure sterile tampon covers the gallbladder bed, instead of the insecure open cavity that remains after classical cholecystectomy*’ [20] is noteworthy as it corresponds with the principles of safety in cholecystectomy, which provide a guide for preventing injuries to the bile ducts of any defined order, including subvesical ones. In 1944, G. Grey Turner (London, UK) described them in this manner [25] (p. 621): 

‘*This (i.e., cystic) duct must be exposed and isolated; however, no clamp or ligature should be applied to what is supposed to be the cystic duct until the common hepatic duct, and the common bile duct are clearly observed. I freely admit that there are occasions when the surgeon cannot be satisfied with this anatomical disposition, but in such circumstances, the proper course is either to be content with a partial cholecystectomy, leaving the portion of the viscus just above the neck of the gallbladder, or cholecystostomy*’.

In the rare event of fibrotic, contracted and partially buried gallbladder when the cystic duct and artery can be safely exposed to surrounding pericholecystic tissues, it is safe to follow the principles of Thorek’s surgery for the gallbladder of which an approximation of the edges of the gallbladder should be discarded.

#### 3.7.2. Cholecystectomy by W. A. McElmoyle

In 1954, W. A. McElmoyle described a technique of cholecystectomy for the difficult gallbladder [31]. Partial removal of the distal part of the hepatic wall—a surgical action that was not new—was the essential feature of this operation [22,31]. McElmoyle did not specify whether this was a partial or subtotal cholecystectomy method. In contrast, McElmoyle description suggests that the method was considered a modification of cholecystectomy for a difficult gallbladder. It entailed the following three primary and two additional components: (1) excision of the peritoneal wall leaving in situ the cystic duct and part of the gallbladder wall, viz. portion of the neck, infundibulum, and body lying above and to the left against the liver bed, which is a hepatic wall from half to three-quarters. (2) sutures around the edge of the remaining gallbladder wall to control bleeding. (3) chemical or electrical destruction of the mucosa of the remnant of the gallbladder. (4) splitting the cystic duct with or without opening the common bile duct. (5) fulguration of the mucosa of the cystic duct to within 5 mm of the common duct. The last two surgical actions are indicated if they are advisable. McElmoyle’s operation should be considered as one of the technical subvariants of a partial cholecystectomy, which encompasses a partial excision of the hepatic wall when possible. 

In addition, W. A. McElmoyle’s unique phrases and terms, such as ‘imaginary line’, ‘dangerous area’, ‘shield to the vulnerable structures’, and ‘protective shield’ were used to highlight the rationale for this operative technique and to draw surgeons’ attention to preventative measures performing cholecystectomy. The term ‘a protective shield’ (a part of the remaining hepatic wall of the gallbladder) alludes to W. D. Haggard’s (Nashville, Tennessee) phrase used in discussing W.L. Estes’ paper in 1938: ‘*The little piece of gallbladder adherent to the bed of the liver is the patient’s protection* [17]’. The ‘nature’s protective barrier’ was another term of a similar meaning used to describe the merits of modified cholecystectomy by L. J. Morse and J. S. Barb, who quoted W.L. Estes’ paper in 1947 [16,28]. Therefore, it is reasonable to eponymise this specific phrase as Estes-Haggard’s protective barrier. 

### 3.8. The Changing Terminology: Subtotal Cholecystectomy

The available literature suggests that this happened in the 1940s and 1950s in the Americas [28,34,35,36,37]. However, it is necessary to admit that the change in terminology was slow. Most surgeons used the traditional term ‘partial cholecystectomy’ until the new impulse was generated in South Africa in 1985 [38]. 

#### 3.8.1. Morse and Barb, 1947: Introduction of the New Term

L. J. Morse and J. S. Barb (New York, NY, USA) introduced the term ‘subtotal cholecystectomy, entirely synonymous with the traditional term ‘partial cholecystectomy’, in 1947 [28]. They stated that subtotal cholecystectomy, as a procedure of inestimable value in the management of acutely ill patients with fulminant cholecystitis, is the removal of the gallbladder to the cystic duct except for that portion attached to the liver. The clear emphasis is that no attempt should be made to isolate the cystic duct or artery in the brawny induration of the hepatoduodenal ligament. Additionally, they advocated cystic duct drainage by inserting a catheter in the lumen to decompress the infected biliary tract.

L. J. Morse and J. S. Barb also stressed that complete cholecystectomy is the operation of choice in treating gallbladder disease. Subtotal cholecystectomy should not be considered to supplant it. Only when complete removal cannot be safely accomplished, a partial cholecystectomy, rather than cholecystostomy, should be present for the surgeon’s consideration as it combines all the advantages of cholecystectomy with none of the limitations of cholecystostomy and escapes the added risk of conventional cholecystectomy.

#### 3.8.2. Bonilla Naar, 1954: Colombia

In 1954, Alfonso Bonilla Naar introduced subtotal cholecystectomy as a new technique for gallbladder surgery (Bogota, Colombia) [34].

#### 3.8.3. Madding, 1955: Subtotal Cholecystectomy as a Modification of Partial Cholecystectomy 

Gordon F. Madding (San Mateo, CA, USA) wrote in 1955 that subtotal cholecystectomy is a conservative gallbladder operation modified by partial cholecystectomy. It should be used when the hazards to the biliary tree are substantial; this implies when gallbladder surgery is conducted in an operative field where all normal anatomy is obscured [35]. Specifically, each incomplete gallbladder excision is partial cholecystectomy, whereas subtotal cholecystectomy is one modified technical variant of partial cholecystectomy. The descriptions of both partial and subtotal cholecystectomies were as follows [35] (pp. 604–605): 

‘*An incision may then be made from the fundus to within 1 cm of the cystic duct and all stones removed, particularly the one creating the obstruction of the cystic duct. No attempt is made to isolate the cystic duct or artery in the presence of the brawny induration in the hepatoduodenal ligament. When, following the removal of all stones, the redundant flaps of the gallbladder are trimmed off at the liver bed attachment, the procedure has been designated in the literature as partial cholecystectomy, an operation first described in 1899. When the gallbladder is dissected in a retrograde fashion from the surface of the liver proper down to within 1 cm of the cystic duct, the method will be referred to as “subtotal cholecystectomy” and is the preferable procedure… A Penrose drain is then placed into the gallbladder stump, which remains; this gallbladder remanent usually not exceeding 1 cm in diameter. The cuff is then closed about the drain with interrupted catgut sutures. Other drains are placed, and these, in turn, are brought out the lateral angle of the wound where a subcostal type of incision has been used*’.

Briefly, a circumferential resection of the gallbladder at the neck level was defined as subtotal cholecystectomy. The closure of the remnant of the gallbladder was another feature of subtotal cholecystectomy. Madding’s description of subtotal cholecystectomy broadened its comprehension. 

#### 3.8.4. Farrow’s Thesis on Subtotal Cholecystectomy, 1958–1959

Charles Durrett Farrow (Miami, FL, USA), who overviewed 13 papers on partial cholecystectomy published between 1926 and 1954, reported results of 24 subtotal cholecystectomies and provided a different definition of subtotal cholecystectomy [36,37]. A standard surgical procedure that implied the removal of the free portion of the gallbladder while a portion attached to the liver bed remains in situ without exposure and ligature of the cystic duct was confirmed to be a subtotal cholecystectomy. However, the section ‘Techniques’ of the first Farrow’s paper [36] shows that the interpretation of the term ‘subtotal cholecystectomy’ may have a broader range because it includes a modification of partial cholecystectomy with exposure and ligature of both cystic artery and duct [22]. Thorek’s operation was the object of Farrow’s subtotal cholecystectomy review [18,19,20].

Another highlight of Farrow’s thesis was an emphasis on the contraindications for subtotal cholecystectomy. The prima-facie examples included carcinoma of the biliary tract, the head of the pancreas, and the duodenum.

### 3.9. New Wave Developing a Conception of Subtotal Cholecystectomy, 1985–1991: Techniques to Secure a Cystic Duct 

A life-threatening complication of cholecystitis and advanced portal hypertension was one of the indications for subtotal cholecystectomy in P.C. Bornman and J. Terblanche’s (Cape Town, South Africa) case series [38]. Their subtotal cholecystectomy technique entailed a resection of the peritoneal wall of the gallbladder, a running suture on the rim of this wall, and identification of cystic duct origin from inside the gallbladder. A piecemeal excision of the peritoneal wall of the gallbladder, starting at Hartmann’s pouch, was the first remarkable point. Next, two techniques were suggested to secure the cystic duct—a ligature around the cystic duct with the help of a probe inside the cystic duct and closure of the internal orifice of the cystic duct from within the gallbladder with a purse string or oversew technique. The latter technique to secure the cystic duct after partial resection of the gallbladder was the second remarkable point of this paper. 

Moshe Schein (Johannesburg, South Africa), in a paper on partial cholecystectomy published in 1991 [41], clarified that the accurate placement of the purse string around the cystic duct opening, as described by Borman and Terblanche, is not satisfactory as the suture tends to tear out in the inflamed and friable tissues [40,43,44]. Therefore, Schein suggested leaving a 1 cm rim of Hartmann’s pouch and buttressing it over the opening of the cystic duct. Additionally, ‘the fundectomy’, another operative variant of partial cholecystectomy, which includes the excision of the fundal portion of the gallbladder with the closure of the gallbladder remnant in continuity with the cystic duct, thus enabling the risk of recurrent gallstone formation, was mentioned in this paper.

### 3.10. Subtotal Cholecystectomy: Khan’s Modification, 1992

T. F. T. Khan’s (Kubang Kerian, Malaysia) modification of subtotal cholecystectomy for the difficult gallbladder is a combination of a longitudinal incision through the peritoneal wall of the gallbladder and circumferential division of its neck, the removal of the gallbladder above the circumferential incision, and the obliteration of the cavity of the neck of the gallbladder with interrupted sutures, obtained at an appropriate distance from the cut edge [43].

It is a clearly illustrated description of partial circumferential resection of the gallbladder with obliteration of the remaining small proximal portion of the gallbladder. It was applied in 43 patients between 1985 and 1990. No adverse events related to this procedure were reported.

### 3.11. First Reports on Laparoscopic Subtotal Cholecystectomy, 1993

In 1993, Amitai Bicker and Benjamin Shtamler (Nahariya, Israel) reported six laparoscopic subtotal cholecystectomies—excision of the peritoneal wall and electrocoagulation of the mucosa of the hepatic wall of the gallbladder [45]. Two of them had macronodular liver cirrhosis. The gallbladder was partially embedded in the liver tissue in three patients. Identification, isolation, and division of the cystic duct and artery were conducted in all patients. Postoperative recovery of all patients was uneventful.

G. Crosthwaite (Glasgow, Scotland) reported the outcomes of five patients and concluded that laparoscopic subtotal cholecystectomy is a safe procedure and can be used in selected patients to avoid conversion to open surgery [47].

### 3.12. The Systematisation of Modalities of Subtotal Cholecystectomy, 1993–2022

Table 1 summarises the systematisation of technical modalities of resections of the gallbladder and the most characteristic types of completion of subtotal cholecystectomy.

#### 3.12.1. Ibrarullah, 1993: Two Modalities of Partial Cholecystectomy

Ibrarullah et al. (Lucknow, India) classified partial cholecystectomy into two subgroups within one series of their operations [46]. Partial cholecystectomy with retained Hartmann’s pouch with or without closure of the internal orifice of the cystic duct by a purse string suture of polyglactin and partial cholecystectomy with retained hepatic wall with the approximation of the free margins of the gallbladder pouch, when feasible, by a continuous absorbable suture were performed in 17 and 12 patients, respectively. Thus, each modality includes two techniques—with or without closing the cystic duct and with or without approximating the remnant of the gallbladder. However, the results of these four techniques were not presented separately. 

#### 3.12.2. Crosthwaite, 1995, and Michalowski, 1998: Laparoscopic Surgery 

The systematisation of the first laparoscopic resections of the peritoneal wall of the gallbladder according to the presence (clipping/ligation) or absence of intervention onto the cystic duct is traced in Crosthwaite’s paper [47]. The cystic duct was closed in three of the five patients. No bile leakage was observed in all of them. 

K. Michalowski’s (Cape Town, South Africa) description, provided 3 years later, was similar with one difference—29 patients constituted a surgical cohort [52]. In 27 patients, the cystic duct was isolated from the surrounding tissues and clipped, ligated, or sutured. In Michalowski’s series, bile leaks occurred in three (11%) patients in whom the cystic duct stump was secured with metal clips. However, the need for open conversion or cholecystostomy was reduced.

#### 3.12.3. Maudar, 1996: Classification of Resections of the Portions of the Gallbladder 

Partial and subtotal cholecystectomies are different operations according to K. K. Maudar’s (Pune, India) interpretation of partial resections of the gallbladder [48]. Partial cholecystectomy is a circumferential excision of the gallbladder at the level of the neck. Mirizzi syndrome type I (five operations within the series) was the indication to perform partial cholecystectomy. Bornman and Terblanche’s operation [38] was classified as subtotal cholecystectomy in the other 21 surgical cases.

#### 3.12.4. Palanivelu, 2006, and Gode, 2014: Three Variants of Subtotal Cholecystectomy 

Chinnasamy Palanivelu (Coimbatore, India) classified difficulties in performing laparoscopic cholecystectomy in patients with liver cirrhosis of Child–Pugh A and B classes under five headings; consequently, he standardised the laparoscopic subtotal cholecystectomy techniques into three variants [56]. The first variant—laparoscopic subtotal cholecystectomy I (viz. LSC I)—entailed resection of the peritoneal wall leaving the high-risk hepatic wall in situ; the remnant mucosa was recommended to be removed in patients with acute cholecystitis or by electrofulguration in those with chronic cholecystitis. 

The second variant—LSC II—is performed in the presence of high-risk hepatic hilum. It requires a circumferential transection of the infundibulum as close to the gallbladder and cystic duct junction as safely possible. As in variant LSC I, the mucosa in the proximal remnant is removed by mucosectomy in patients with acute cholecystitis and by electrofulguration in those with chronic cholecystitis. The remnant of the gallbladder is closed by suture. A combination of LSC I and LSC II variants is recommended in patients with high-risk gallbladder beds and hepatic hilum. This new variant was named LSC III. The remaining hepatic wall and closure of the remnant of the gallbladder are the features of this variant. In the Palanivelu series of 265 operations, LSC I, II and III were performed in 62 (23.4%), 102 (38.5%), and 42 (15.9%) patients, respectively. 

In 2014, to reduce operating time, the rates of conversion to laparotomy, bleeding, injury to biliary ducts, and duration of postoperative hospital stay, the C. Palanivelu and Dilip Gode (Wardha, India) group of surgeons extended the indications for laparoscopic subtotal cholecystectomy across all age groups of the cohort of 661 patients and added component—dissection, clipping and transection of cystic duct and artery—to variant LSC I [83]. Therefore, LSC I was subcategorised into two subvariants—LSC I without cystic duct closure [56] and LCS I with cystic duct closure [83].

#### 3.12.5. Henneman, 2013: Four Methods of Partial Cholecystectomy

Daniel Henneman reviewed 15 papers on partial cholecystectomy, published between 1993 and 2010, and distinguished four methods of partial cholecystectomy used by the studies’ authors [159]. Excision of the peritoneal wall of the gallbladder (viz. without closure of the remaining stump), leaving the drain was defined as Method A. Excision of the peritoneal wall of the gallbladder with the closure of the remaining stump was defined as Method B. Partial cholecystectomy method C entailed a distal resection of both walls of the gallbladder from the transection line at the level of its neck or Hartmann’s pouch and closure of the remaining proximal portion of the gallbladder; without drainage. Furthermore, when the remaining proximal portion of the gallbladder was left open (after removing the distal one) with a drain close to it, it was attributed to Method D. Electrical or laser coagulation of the mucosa of the remnant portion of the gallbladder was considered a matter of secondary importance within this classification of the partial cholecystectomy methods. Notably, Henneman’s review highlighted the cause and pattern of postoperative bile leak regarding the application of the partial cholecystectomy method: bile leak occurred in 5.6% of the 321 patients with closed cystic duct compared with 16.3% of the 295 with an open cystic duct. Regarding this, Method A was the most burdensome. 

#### 3.12.6. Strasberg, 2016, LeCompte 2020: Fenestrating and Reconstituting Subtypes

Steven M. Strasberg et al., in a special article published in 2016, expressed the opinion to discard the term ‘partial cholecystectomy’ because, first, *‘it is quantitatively a vague term that can mean the removal of a small or large part of the organ’*. Second, ‘*it is a source of confusion that two terms are used for the same procedure’* [160]. A classification of gallbladder operations where only a part of this organ is removed, called partial cholecystectomy, was proposed. It included two components—subtotal cholecystectomy and fundectomy. Partial resection of approximately 75% or more of the gallbladder was defined as subtotal cholecystectomy. Resection of the top half (50%) or less of the gallbladder implied a fundectomy. Moreover, it was emphasised that the term ‘partial cholecystectomy’ could be eliminated, fully referencing the partial excisions of the gallbladder to any extent.

Furthermore, another source of confusion—limited nomenclature relating to gallbladder remnants—was mentioned in the same article [160]. Therefore, new terms were introduced to indicate whether a particular technique leaves a remnant gallbladder. Two of them were subtotal fenestrating cholecystectomy and subtotal reconstituting cholecystectomy. Surprisingly, two different words—subtypes and types—were used to characterise subtotal cholecystectomy in this respect.

Regarding subtotal fenestrating cholecystectomy, it materialised into four standard forms of technical execution of the operation—without or with partial excision of the hepatic wall, and without or with the closure of the internal orifice of the cystic duct with a purse-string suture—in both papers [126,160]. Subtotal reconstituting cholecystectomy covered two technical modalities—without or with partial removal of the hepatic wall of the gallbladder. 

Four years later, in 2020, LeCompte reported experience and outcomes with subtotal cholecystectomy in the years immediately preceding adoption and since the adoption of this theory [126]. Regarding short-term postoperative outcomes, eight (11.3%) of the 71 patients required a secondary surgery—either endoscopic retrograde cholangiopancreatography or interventional drainage placement—due to bile leakage. At a mean follow-up of approximately 1 year, no patient returned with recurrent symptoms.

#### 3.12.7. Tokyo Guidelines, 2007–2018: Acute Cholecystitis, Severity Grades, and the Surgical Alternatives

The Tokyo guidelines (TG), which have achieved an international consensus, were first published in 2007 [161,162,163]. They provide knowledge and recommendations on various aspects of acute cholecystitis and cholangitis, including the management of these biliary diseases. Since 2007, all three versions of TG—TG07, TG13, and TG18—have been widely cited (https://scholar.google.com (accessed on 28 January 2023), search phrase ‘Tokyo guidelines’). 

Subtotal (partial) cholecystectomy was overviewed in TG18. K. Okamoto et al. briefly characterised a subtotal cholecystectomy as a surgical alternative to laparoscopic cholecystectomy that should be performed with particular care because of the difficulty associated with operating on patients with grade II (moderate) acute cholecystitis in an advanced surgical centre [164]. The logic of the second recommendation of this paper dictates that the same switch should also be considered in difficult surgical situations related to grade III (severe) acute cholecystitis in those patients who were preoperatively considered suitable for gallbladder resectional surgery.

G. Wakabayashi et al. pointed out that the differences between the terms ‘partial cholecystectomy’ and ‘subtotal cholecystectomy’ need to be clarified [165]. Although Strasberg’s definitions of the optimal procedure in difficult operative conditions (see Section 3.12.6) were described in this TG18 paper [165], no distinction between a subtotal and a partial cholecystectomy was provided. On the contrary, in the proposal made for avoiding vasculo-biliary injury based on the Delphi consensus (2017) [166], this paper indicated that the words ‘subtotal’ (preferable) and ‘partial’ (less preferable) can be used as synonyms [165]. 

#### 3.12.8. Lunevicius, 2020: Subtotal Cholecystectomy Resectional Variants and Subvariants

Subtotal cholecystectomy is a resection of the gallbladder portion, of any size, according to the line of thought arising from this paper [167]. To characterise the extent of the laparoscopic resections quantitatively and the surgical techniques used, four technical variants—STC-1 (80%), STC-2 (18%), STC-3 (1.6%), and STC-4—and five subvariants for the first two variants (STC-1A, STC-1B, STC-1C, and STC-2A, STC-2B) were proposed. 

According to this classification, STC-1 implies a circular excision of a considerable portion of the gallbladder at the level of the infundibulum or neck. STC-2A subvariant entails the removal of the peritoneal wall of the gallbladder; STC-2B subvariant includes external closure of the cystic duct, removal of the peritoneal wall of the gallbladder, and, when possible, partial removal of the hepatic wall. STC-3 counts for a fundectomy. STC-4 is a wedge (viz., minimalistic) resection of the wall of the gallbladder. 

The gallbladder resection variant and the type of completion of the resectional operation are two different categories. The fenestrating and reconstituting types of subtotal cholecystectomy completion were examined in the paper, given the trend in the literature published between 2016 and 2019. The same classification was used to characterise the technical variants of subtotal cholecystectomies performed via laparotomy [7,137]. 

#### 3.12.9. Purzner 2019, Deng 2022: Five Subtypes of Laparoscopic Subtotal Cholecystectomy

Roderick H. Purzner [113] and Shirley X. Deng et al. [143] from Toronto (Canada) described a classification incorporating five subtypes of laparoscopic subtotal cholecystectomy. Circumferential amputation of the gallbladder with a closure of the remnant portion of the gallbladder (<1 cm) or internal orifice of the cystic duct was named LSC 1A; without a closure of one of these two anatomical structures—as LSC 2A. A subtype of subtotal cholecystectomy entailing an opening of the peritoneal wall of the gallbladder via ‘T’ form incision, removal of this wall and a suture-closure of the cuff against the hepatic wall of the gallbladder was defined as LSC 1B. When an operation included the first two of the three surgical components of the LSC 1B, leaving the cystic duct open, it was defined as LSC 2B. In Deng’s paper [143], LSC subtypes 1A and 1B (65.2% and 4.3% of the cases) were attributed to reconstituting subtotal cholecystectomy, LSC subtypes 2A and 2B (15.2% and 8.7%)—to fenestrating type of subtotal cholecystectomy. 

A new subtype—LSC 3, characterised as a damage-control fenestrating subtotal cholecystectomy—was introduced [113,143]. It was recommended in the setting of extensive adhesions between the intestine and the peritoneal wall of the gallbladder. Two operative components—a fundectomy and a bilateral longitudinal split of the gallbladder above both margins of the cystic plate to open the gallbladder like a clamshell—constituted the LSC 3 (6.5%) [143].

Therefore, Deng’s paper [143] made the vague and unprecise terms ‘fenestrating’ and ‘reconstituting’ [126,160] accurate. Subtotal cholecystectomy with an open cystic duct within the peritoneal cavity was characterised as subtotal fenestrating cholecystectomy. Subtotal cholecystectomy with a closed biliary tract (in the form of an internal suture to the cystic duct or closure of the gallbladder remnant) was characterised as subtotal reconstituting cholecystectomy. Additionally, Deng et al. recognised that fundectomy is a subtype of subtotal cholecystectomy. However, no space within the classification was given for a subtype which entails external closure of the cystic duct and resection of the peritoneal wall of the gallbladder.

**Table 1 jcm-12-01230-t001:** The systematisation of partial resections of the gallbladder, by the individual article: types, variants, methods, and procedure completion.

Author, Year	Resection		Removal of the Gallbladder Wall		Extent of Resection *	Cystic Duct	Gallbladder Remnant	Procedure Completion, by Key Characteristic
	Type/Variant/ Method	Subtype/ Subvariant	Peritoneal	Hepatic				
Ibrarullah, 1993 [46]	PC with retained Hartmann’s pouch (*n* = 17)	NA	Yes	Yes	80–90% or more	Open	Closed	Closed-tract STC
	PC with retained posterior wall (*n* = 12)	NA	Yes	No	75%	Closed	Open	Closed-tract STC
Crosthwaite, 1993 [47]	STC, open CD (*n* = 3)	NA	Removal	In situ	75%	Open	Open	Open-tract STC
	STC, closed CD (*n* = 2)	NA	Removal	In situ	75%	Closed	Open	Closed-tract STC
Maudar, 1996 [48]	PC (*n* = 5)	NA	Removal	Removal	80–90% or more	Open	Closed	Closed-tract STC
	STC (*n* = 21)	NA	Removal	In situ	75% or less	NA	NA	NA
Michalowski, 1996 [52]	STC with isolation and division of CD (*n* = 27)	NA	Removal	In situ	75%	Closed	Open	Closed-tract STC
	STC without isolation and division of CD (*n* = 2)	NA	Removal	In situ	75%	Open	Open	Open-tract STC
Palanivelu, 2006 [56]	LSC I (*n* = 62)	NA	Yes	No	75%	Open	Open	Open-tract STC
	LSC II (102)	NA	Yes	Yes	90%	Open	Closed	Closed-tract STC
	LSC III (*n* = 42)	NA	Yes	No	75%	Open	Closed	Closed-tract STC
Gode, 2014 [83]	LSC I (*n* = 48)	NA	Yes	No	75%	Closed	Closed	Closed-tract STC
	LSC II (*n* = 591)	NA	Yes	Yes	90%	Open	Closed	Closed-tract STC
	LSC III (*n* = 22)	NA	Yes	No	75%	Open	Closed	Closed-tract STC
Henneman, 2013 ** [159]	Method A	NA	Yes	No	75%	Open	Open	Open-tract STC
	Method B	NA	Yes	No	75%	Open or closed	Closed	Closed-tract STC
	Method C	NA	Yes	Yes	90%	Open	Closed	Closed-tract STC
	Method D	NA	Yes	Yes	90%	Open	Open	Open-tract STC
Strasberg, 2016 LeCompte 2020 [126,160]	NA	Reconstituting	Yes	Yes	90%	Open	Closed	Closed-tract STC
	NA	Fenestrating	Yes	No	75%	Open or closed	Opened	Open-tract STC Closed-tract STC
Lunevicius, 2021 [167]	STC-1	STC-1A	Removal	Removal	80–90% or more	Open	Open or Closed	Open-tract STC Closed-tract STC
		STC-1B	Removal	Removal	80–90% or more	Open	Open or Closed	Open-tract STC Closed-tract STC
		STC-1C	Removal	Removal	80–90% or more	Open	Open or Closed	Open-tract STC Closed-tract STC
	STC-2	STC-2A	Removal	In situ	75%	Open	Open	Open-tract STC
		STC-2B	Removal	Partial	75% or more	Closed	Open	Closed-tract STC
	STC-3 (fundectomy)	NA	Fundus removal	NA	Fundectomy	Open	Open or Closed	Open-tract STC Closed-tract STC
	STC-4	NA	Removal	In situ	Minimal	Open	Open	Open-tract STC
Purzner, 2019 Deng, 2022 [113,143]	Reconstituting	LSC 1A	Excision	Excision	80–90% or more	Open	Closed	Closed-tract STC
	Reconstituting	LSC 1B	Excision	In situ	75%	Open	Closed	Closed-tract STC
	Fenestrating	LSC 2A	Excision	Excision	80–90% or more	Open	Open	Open-tract STC
	Fenestrating	LSC 2B	Excision	In situ	75%	Open	Open	Open-tract STC
	Fenestrating	LSC 3	Bilateral split	In situ	NA	Open	Open	Open-tract STC

Asterisks: * It is the approximate extent. ** review paper. Abbreviations: PC—partial cholecystectomy, *n*—number of operations performed, STC—subtotal cholecystectomy, LSC—laparoscopic subtotal cholecystectomy. NA—no information or unclear description of the surgical method. Explanation, the last column: Procedure completion type, by key characteristic, indicates whether the cystic duct or the gallbladder remnant was closed (closed-tract STC = subtotal closed-tract cholecystectomy) or not (open-tract STC = subtotal open-tract cholecystectomy).

## 4. Discussion

### 4.1. Historical Evolution and Current Trends

The evolution of the resection of part of the gallbladder began in the last decade of the 19th century, according to available literature sources of which Kehr and Mayo’s names are frequently quoted [17,35,36,160,168]. The technique of partial resection of the gallbladder leaving the hepatic wall in situ was well described in the 3rd and 4th decades of the 20th century by A. J. Bengolea, A. Zabala, T. De Martel, V. Pauchet, E. D. Martin, W. D. Gatch, W. L. Estes, H. P. Ritchie, and J. McKenty, and acknowledged by contemporaries. Estes emphasised this operation as a ‘partial cholecystectomy’ [15]. 

The term ‘subtotal cholecystectomy’ as an alternative to the term ‘partial cholecystectomy was first introduced in 1947 by Morse and Barb [28], popularised by Madding [35] and Farrow [36,37] and revitalised by Bornman and Terblanche in 1985 [38], has been dominant in the medical literature since 2014. It replaced the term ‘partial cholecystectomy’ in 2017. Figure 2 demonstrates this trend. Fundectomy is also considered one of the variants [137,167] or subtypes [143] of subtotal cholecystectomy. Therefore, ‘subtotal cholecystectomy’ became equivalent to ‘partial cholecystectomy’, maximally minimising the field for speculations on differences between definitions of partial and subtotal cholecystectomy [4,109,150,169]. This shift in terms complies with an Office of Population Censuses and Surveys Classification of Interventions and Procedures Fourth Revision principle to enfold partial resections of the gallbladder under one nomenclatural term, hence reducing the bias in statistical interpretations [170]. Currently, it is a ‘partial cholecystectomy’ [170].

A categorisation of subtotal cholecystectomy into fenestrating and reconstituting types is another apparent trend in the medical literature [126,160,171,172]. Additionally, two recent systematic reviews compared the clinical outcomes between the subgroups based on these categories [8,172]. However, it should be acknowledged that the interpretation of the definitions (viz., fenestrating and reconstituting) in these two independently conducted reviews only partially corresponded with the original definitions of both subtotal cholecystectomy types. 

Two other additional points should also be noted. First, a subvariant of subtotal cholecystectomy with isolation and division of the cystic duct (STC-2B, [167]) is beyond the scope of classification based on ‘fenestration–reconstitution’ terminology [126,160]. Second, a recently proposed damage-control LSC 3 subtype of laparoscopic subtotal cholecystectomy, which is a fundectomy with a bilateral split of the peritoneal wall of the gallbladder [143], contradicts Strasberg’s unique proposal [160], as fundectomy, a priori, was not a part of it. Therefore, these are examples of why the terms—subtotal fenestrating cholecystectomy and subtotal reconstituting cholecystectomy—should be discarded. 

The severity of inflammation of the gallbladder, liver cirrhosis, and pericholecystic inflammatory changes (for example, the presence of chronic inflammatory mass) determine the site and extent of resection of the gallbladder. Therefore, a trend of systematising the extent of resection is real [56,83,143]. It corresponds well with the long evolution of subtotal cholecystectomy, as presented in Table 2, which summarises historical descriptions and case series of less-than-complete cholecystectomy in chronological order, from 1898 to 1999. For example, this table shows that the excision of the peritonealised wall of the gallbladder was the essence of partial cholecystectomy for a long time. However, McElmoyle in 1954 [31], Madding in 1955 [35], and Khan in 1992 [43] emphasised the necessity, when possible, to excise the part of the hepatic wall of the gallbladder.

Another significant trend for performing limited cholecystectomy is demonstrated in Table 2. The authors reported closure of the cystic duct or the remaining gallbladder in 60% of studies (*n* = 24). In addition, the systematic reviews reported a similar or higher proportion of studies with the closure of the cystic duct and gallbladder remnant, 62% [172] and 75% [8]. Both these reviews have demonstrated that the clinical outcomes associated with the effects of bile leakage favoured the patient subgroup with the closed remnant of the gallbladder or cystic duct.

### 4.2. The Outline of a Conception of Subtotal Cholecystectomy

Subtotal cholecystectomy is a principle-guided multimodal surgical procedure. It aims to prevent the morbidity and mortality burden from injuries to bile ducts, major prehepatic and intrahepatic blood vessels or the gastrointestinal tract in extraordinary surgical situations that differ in origin and nature. The basic principle of subtotal cholecystectomy is to resect a part of the diseased gallbladder with its contents, leaving the rest of this organ (the reason for subtotal cholecystectomy) in situ. The second subtotal cholecystectomy principle, which is the prevention of intracorporeal or extracorporeal bile leak, can be applied when post-resectional surgical circumstances are favourable to prevent bile leakage via the cystic duct and gallbladder remnant.

Therefore, subtotal cholecystectomy is a single-stage or two-stage surgical procedure. Single-stage operation includes resection of the gallbladder. The extent of partial resections varies from approximately 90–95% when a part of the gallbladder neck remains in situ, 75% when the whole hepatic wall is unresectable and left in its anatomical site as Estes-Haggard protective barrier, and 10–30% when a visible part of the fundus or whole fundus is resected (due to having undetachable surrounding chronic pericholecystic inflammatory mass). The extent of resection should be described in the operation note and/or specified via available systems aimed to specify the margins of the gallbladder resection. 

Two-stage subtotal cholecystectomy entails partial resection of the gallbladder and closure of the cholecysto-cystico segment of the biliary tract at any anatomical point of it. Closing the gallbladder remnant is a more prevalent method of completing a two-stage subtotal cholecystectomy; additionally, vice versa, closing the cystic duct is a less prevalent method of completing this procedure. The closure method of these anatomical structures depends on the surgeon’s judgement and choice. 

The two-type classification of subtotal cholecystectomy—fenestrating and reconstituting—lacks specificity, particularly the word ‘fenestrating’. The word ‘reconstituting’ is unusual, too, as the obliteration of the cavity of the remaining small proximal portion of the gallbladder by individual or continuous sutures is an achievable surgical action. 

‘Subtotal open-tract cholecystectomy’ and ‘subtotal closed-tract cholecystectomy’ are alternative terms to characterise the type of completion of subtotal cholecystectomy. They are distinct, mutually exclusive, and consistent with the line of thought connecting most publications published within the last 124 years.

### 4.3. Limitations of the Review and Area That Needs Further Research

Some text from the articles were not obtained despite intensive efforts (see details in Table 2). Also, a gap in the literature sources between 1960 and 1980 is apparent. The method used in this review to identify and select literature on partial resections of the gallbladder yielded no results for these two decades. 

The element of subjectivity in evaluating historical publications and technical details is another possible limitation of this review, despite multiple rounds of analyses. Independent assessment of the content of these historical papers by a group of dedicated researchers, scientists and clinicians is a way to minimise the potential bias. 

Only case series with at least five patients, published between 1985 and 2022, were considered for this review, as it was a priori decided that analysis of larger cohorts of patients adds more value to subtotal cholecystectomy conception than individual case reports [7,137]. On the other hand, the analysis of the numerous case reports on limited resections of the gallbladder for benign gallbladder disease should be regarded as an area that warrants further analytical work [173]. For example, these case reports probably include descriptions of unusual and rare technical manoeuvres for gallbladder resections. Systemisation of them would ensure an added value to subtotal cholecystectomy practise—open, laparoscopic, or robot-assisted. 

## 5. Conclusions

This study described the evolution of subtotal cholecystectomy. The names of surgeons are appreciated as objectively as the availability of the historic papers allowed. This review provides an explanatory basis for new eponyms. ‘Estes-Haggard’s protective barrier’ (a hepatic wall of the gallbladder left in situ) is one of them. Further terms of a categorical origin for the characterisation of completion of subtotal cholecystectomy (open-tract and closed-tract) are introduced.

## Figures and Tables

**Figure 1 jcm-12-01230-f001:**
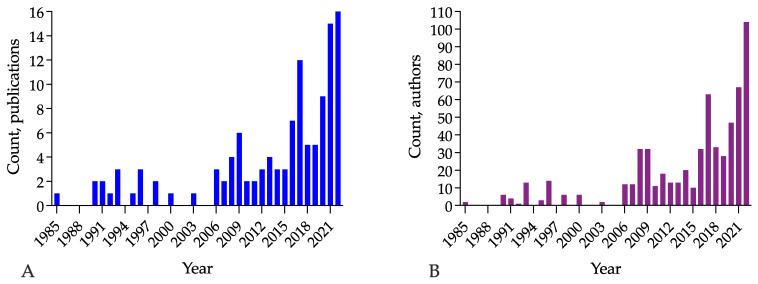
Temporal changes in the number of publications on partial resections of the gallbladder and authors of these articles, from 1985 to 2022. Changes are shown in the number of articles published per year (**A**) and the number of authors by year (**B**). Columns of Figure 1A show the 1600% increase in publications between 1985 (*n* = 1) and 2022 (*n* = 16). A steady 533% increase in articles has been seen since 2006 (*n* = 3); sixteen articles were published in 2022. Columns of Figure 1B show the 5200% increase in the number of authors between 1985 (*n* = 2) and 2022 (*n* = 104). They also demonstrate an 867% increase in authors writing scientific papers on partial resections of the gallbladder (*n* = 12 in 2006, *n* = 104 in 2022).

**Figure 2 jcm-12-01230-f002:**
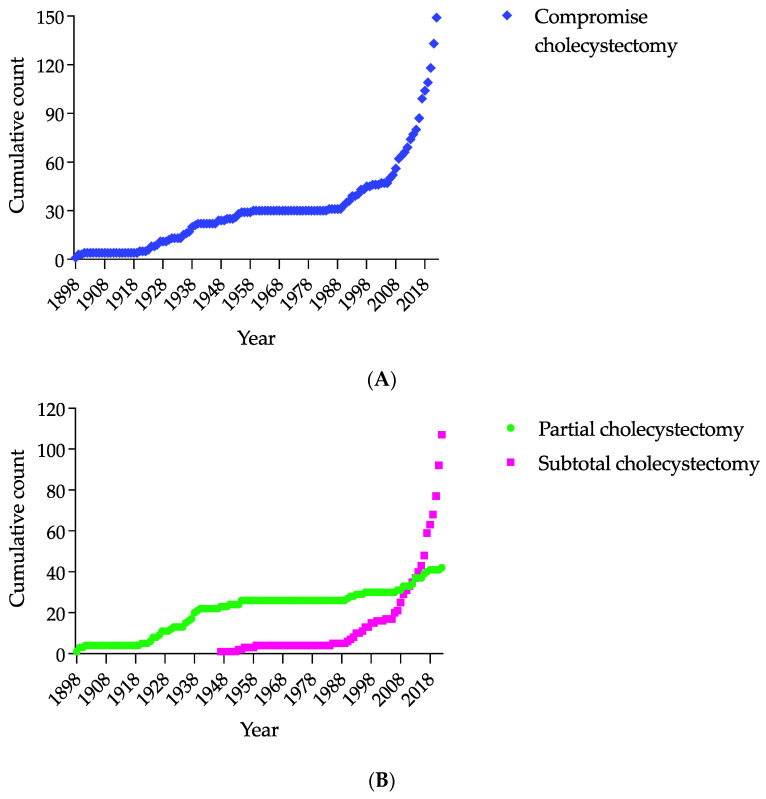
Cumulative annual aggregates of articles on partial resections of the gallbladder, overall and based on the term used, from 1898 to 2022. The dynamics in the cumulative number of articles on these resections, also known as compromise cholecystectomies (**A**) and the cumulative number of articles on operations named partial cholecystectomy and subtotal cholecystectomy (**B**) are shown. Figure 2A demonstrates a sharp rise in the aggregative line since 2008, when 56 articles on less-than-complete cholecystectomies were published. Figure 2B highlights the dominance of the articles with the term ‘subtotal cholecystectomy’ since 2014. In precise terms, 37 articles on partial cholecystectomies and 40 on subtotal cholecystectomies were known by 2014. Eight years later, in 2022, 42 articles on partial cholecystectomies and 107 articles on subtotal cholecystectomies were included in the cumulative count.

**Table 2 jcm-12-01230-t002:** The terms used to describe the resection of a part of the gallbladder in the historical publications and the type of operation completion, in chronological order, from 1898 to 1999.

Year	Author	Cases	Term	Resected Walls		Closure		Completion	
				Peritoneal	Hepatic	CD	Remnant	Open	Closed
1898	Kehr [11]	2	None	+	+	−	+	−	+
1899	Mayo [12]	3	None	+	+	−	+	−	+
1899	Mayo [12]	3	None	NA	NA	−	+	−	+
1901	Stevens [13]	1	None	+	+	−	+	−	+
1920	Bengolea [17] *	NA	PC	NA	NA	−	−	−	−
1923	De Martel [17] *	2	None	+	−	+	−	−	+
1924	Zabala [17] *	5	PC	+	−	−	−	+	−
1924	Pauchet [17] *	NA	PC	+	−	−	−	+	−
1926	Martin [14]	8	None	+	−	−	−	+	−
1927	Gatch [17] *	NA	PC	+	−	−	−	+	−
1927	Zimmerman [17] *	NA	PC	+	−	−	−	+	−
1930	Haggard [17] *	NA	PC	+	−	−	+	−	+
1931	Estes [15]	7	PC	+	−	−	−	+	−
1933	Judd [21]	149	PC	+	−	−	−	+	−
1935	McKenty [17] *	NA	PC	+	−	+	−	−	+
1936	Thorek [18]	201	ESC	+	−	+	−	−	+
1937	Ritchie [23]	16	None	+	−	−	+	−	+
1938	McKenty [22]	30	PC	+	−	−	−	+	−
1938	Estes [16]	48	PC	+	−	−	−	+	−
1938	Thorek [19]	342	ESC	+	−	+	−	−	+
1939	Bailey [26]	129	ESC	+	−	+	−	−	+
1940	Gurd [27]	6	PC	+	−	−	−	+	−
1947	Morse [28]	2	STC	+	−	−	−	+	−
1947	Love [29]	129	ESC	+	−	+	−	−	+
1950	Lerner [30]	2	PC	+	−	−	−	+	−
1953	Meyer [36] **	28	STC	+	−	−	−	+	−
1954	Thorek [20]	NA	ESC	+	−	+	−	−	+
1954	McElmoyle [31]	23	C	+	+	−	−	+	−
1955	Madding [35]	4	STC	+	+	−	+	−	+
1959	Farrow [37]	24	STC	+	−	−	−	+	−
1985	Bornman [38]	18	STC	+	−	+	−	−	+
1990	Douglas [39]	11	PC	+	−	+	−	−	+
1991	Schein [41]	16	PC	+	−	+	−	−	+
1991	Cottier [42]	11	STC	+	−	+	−	−	+
1992	Khan [43]	43	STC	+	+	−	+	−	+
1993	Schein [44]	23	STC	+	−	+	−	−	+
1993	Bickel [45]	6	STC	+	−	+	+	−	+
1996	Subramaniasivam [49]	15	PC	+	−	+	−	−	+
1996	Katsohis [50]	34	STC	+	−	+	−	−	+
1998	Ranson [51]	8	STC	+	−	+	−	−	+

Abbreviations: NA—no information available from the primary or secondary source. CD—cystic duct. PC—partial cholecystectomy. STC—subtotal cholecystectomy. ESC—electrosurgical cholecystectomy or electrosurgical obliteration of the gallbladder (Thorek’s operation). C—cholecystectomy. Asterisks: * Secondary source (Estes, Arch Surg 1938 [17]). ** Secondary source (Farrow, Journal A.O.A. 1958 [36]). Symbols: + Performed. − Not performed. Explanations: Four publications (Ibrarullah, 1993 [46], Crosthwaite, 1995 [47], Maudar, 1996 [48], Michalowski, 1996 [52]) were excluded from this table, for they are shown in Table 1. Completion type indicates (the last two columns) whether the cystic duct or the gallbladder remnant was closed (subtotal closed-tract cholecystectomy) or not (subtotal open-tract cholecystectomy).

## Data Availability

Not applicable.

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
