# Peer review of "Review of the Literature on Partial Resections of the Gallbladder, 1898–2022: The Outline of the Conception of Subtotal Cholecystectomy and a Suggestion to Use the Terms ‘Subtotal Open-Tract Cholecystectomy’ and ‘Subtotal Closed-Tract Cholecystectomy’"

_jcm, 2023, doi:10.3390/jcm12031230_

Round 1

Reviewer 1 Report

This manuscript is very useful to understand the partial resections of gallbladder in acute cholecystitis.

References are very old and I congratulated the author for finding them.

I really appreciate the effort made by the author for this review.

The review appears very long, it is organized an it is not difficult to understand.

Author Response

Response to Reviewer Comments

Point 1: English language and style are fine/minor spell check required  Response 1: Thank you for your feedback. The text in the manuscript has been checked by a language academician.

Reviewer 2 Report

The authors performed a narrative review on a topic of relevance. This is a literature review on partial cholecystectomy dating back to 1898.

The review is an exhaustive review on a topic that is of historical and current interest. Although there are no significant new data on this topic, the authors have presented a well-organized historical account of this interesting topic to the readers.

The paper is organized into sections with each of the subsections in the results reflecting a new time period or a new approach. Each of the sections documented the limitations in the papers that different authors reported in each period.

Overall, an interesting paper that will serve as a reference point for readers interested in this not so common procedure.

Author Response

Response to Reviewer Comments

Point 1: English language and style are fine/minor spell check required.  Response 1: Thank you for your feedback. The text in the manuscript has been checked by a language academician.

Reviewer 3 Report

This is a detailed and interesting historical review of an intervention that, despite its age, is still being performed by experienced surgeons to resolve an entity that has been called "technically difficult cholecystectomy" or "hostile gallbladder", whose main objective is to reduce the morbidity and mortality of acute or scleroatrophic cholecystitis and as prevention of inadvertent injury to the main bile duct.

Initially, there was great controversy about the main techniques: partial cholecystectomy and subtotal cholecystectomy, although subtotal cholecystectomy is currently considered the most feasible and the one that offers the best results.

The presentation is orderly, very well documented and complete, and the writing seems appropriate at all times.

Bibliographic citations are appropriate.

I have no problem with it being approved for publication, without modification

Author Response

(The authors gave the same response as above.)

Reviewer 4 Report

I would like to thank the editor for letting me review this interesting and extensive study.

My only comment is that 'Tokyo Guidelines', which have proposed a recent and international consensus on subtotal cholecystectomy should be mentioned and discussed.

Author Response

Response to Reviewer Comments

Point 1: English language and style are fine/minor spell check required.

Response 1: Thank you for your feedback. The text in the manuscript has been checked by a language academician.

Point 2: My only comment is that 'Tokyo Guidelines', which have proposed a recent and international consensus on subtotal cholecystectomy, should be mentioned and discussed.

Response 2: Thank you very much for this suggestion. I have downloaded and thoroughly analysed 23 TG (TG07, TG13, TG18) series papers and have now included Section 3.12.7 into the main text with 6 new references (161–166). Of these references, K. Okamoto and G. Wakabayashi's papers discuss subtotal (partial) cholecystectomy. The following addition has been made:

3.12.7 Tokyo guidelines, 2007–2018: acute cholecystitis, severity grades, and the surgical alternatives

The Tokyo guidelines (TG), which have achieved an international consensus, were first published in 2007 [161–163]. They provide knowledge and recommendations on various aspects of acute cholecystitis and cholangitis, including the management of these biliary diseases. Since 2007, all three versions of TG – TG07, TG13, and TG18 – have been widely cited (https://scholar.google.com, search phrase ‘Tokyo guidelines’).

Subtotal (partial) cholecystectomy was overviewed in TG18. K. Okamoto et al. briefly characterised a subtotal cholecystectomyas a surgical alternative to laparoscopic cholecystectomy that should be performed with particular care because of the difficulty associated with operating on patients with grade II (moderate) acute cholecystitis in an advanced surgical centre [164]. The logic of the second recommendation of this paper dictates that the same switch should also be considered in difficult surgical situations related to grade III (severe) acute cholecystitis in those patients who were preoperatively considered suitable for gallbladder resectional surgery.

G. Wakabayashi et al. pointed out that the differences between the terms ‘partial cholecystectomy’ and ‘subtotal cholecystectomy’ need to be clarified [165]. Although Strasberg’s definitions of the optimal procedure in difficult operative conditions (see section 3.12.6) were described in this TG18 paper [165], no distinction between a subtotal and a partial cholecystectomy was provided. On the contrary, in the proposal made for avoiding vasculo-biliary injury based on the Delphi consensus (2017) [166], this paper indicated that the words ‘subtotal’ (preferable) and ‘partial’ (less preferable) can be used as synonyms [165].

References:

  • Takada, T.; Kawarada, Y.; Nimura, Y.; Yoshida, M.; Mayumi, T.; Sekimoto, M.; Miura, F.; Wada, K.; Hirota, M.; Yamashita, Y.; et al. Background: Tokyo Guidelines for the management of acute cholangitis and cholecystitis. Hepatobiliary Pancreat. Surg.2007, 14, 1–10. DOI: 10.1007/s00534-006-1150-0.
  • Mayumi, T.; Takada, T.; Kawarada, Y.; Nimura, Y.; Yoshida, M.; Sekimoto, M.; Miura, F.; Wada, K.; Hirota, M.; Yamashita, Y.; et al. Results of the Tokyo Consensus Meeting Tokyo Guidelines. Hepatobiliary Pancreat. Surg. 2007, 14, 114–121. DOI: 10.1007/s00534-006-1163-8.
  • Yamashita, Y.; Takada, T.; Kawarada, Y.; Nimura, Y.; Hirota, M.; Miura, F.; Mayumi, T.; Yoshida, M.; Strasberg, S.; Pitt, H.A.; et al. Surgical treatment of patients with acute cholecystitis: Tokyo Guidelines. Hepatobiliary Pancreat. Surg. 2007, 14, 91–97. DOI: 10.1007/s00534-006-1161-x.
  • Okamoto, K.; Suzuki, K.; Takada, T.; Strasberg, S.M.; Asbun, H.J.; Endo, I.; Iwashita, Y.; Hibi, T.; Pitt, H.A.; Umezawa, A.; et al. Tokyo Guidelines 2018: flowchart for the management of acute cholecystitis. Hepatobiliary Pancreat. Sci. 2018, 25, 55–72. DOI: 10.1002/jhbp.516. Erratum in: J. Hepatobiliary Pancreat. Sci. 2019, 26, 534.
  • Wakabayashi, G.; Iwashita, Y.; Hibi, T.; Takada, T.; Strasberg, S.M.; Asbun, H.J.; Endo, I.; Umezawa, A.; Asai, K.; Suzuki, K.; et al. Tokyo Guidelines 2018: surgical management of acute cholecystitis: safe steps in laparoscopic cholecystectomy for acute cholecystitis (with videos). J Hepatobiliary Pancreat Sci. 2018, 25, 73–86. DOI: 10.1002/jhbp.517.
  • Iwashita, Y.; Hibi, T.; Ohyama, T.; Umezawa, A.; Takada, T.; Strasberg, S.M.; Asbun, H.J.; Pitt, H.A.; Han, H.S.; Hwang, T.L.; et al. Delphi consensus on bile duct injuries during laparoscopic cholecystectomy: an evolutionary cul-de-sac or the birth pangs of a new technical framework? Hepatobiliary Pancreat. Sci. 2017, 24, 591–602. DOI: 10.1002/jhbp.503.